# Identification of potent and selective *N*-myristoyltransferase inhibitors of *Plasmodium vivax* liver stage hypnozoites and schizonts

Diego Rodríguez-Hernández[1,7,9], Kamalakannan Vijayan[2,8,9], Rachael Zigweid[2,3], Michael K. Fenwick [2,3], Banumathi Sankaran [4], Wanlapa Roobsoong [5], Jetsumon Sattabongkot [5], Elizabeth K. K. Glennon[2], Peter J. Myler [2,3,6], Per Sunnerhagen [1], Bart L. Staker[2,3], Alexis Kaushansky [2,6] ✉ & Morten Grøtli [1] ✉

Drugs targeting multiple stages of the *Plasmodium vivax* life cycle are needed to reduce the health and economic burdens caused by malaria worldwide. *N*-myristoyltransferase (NMT) is an essential eukaryotic enzyme and a validated drug target for combating malaria. However, previous *Pv*NMT inhibitors have failed due to their low selectivity over human NMTs. Herein, we apply a structure-guided hybridization approach combining chemical moieties of previously reported NMT inhibitors to develop the next generation of *Pv*NMT inhibitors. A high-resolution crystal structure of *Pv*NMT bound to a representative selective hybrid compound reveals a unique binding site architecture that includes a selective conformation of a key tyrosine residue. The hybridized compounds significantly decrease *P. falciparum* blood-stage parasite load and consistently exhibit dose-dependent inhibition of *P. vivax* liver stage schizonts and hypnozoites. Our data demonstrate that hybridized NMT inhibitors can be multistage antimalarials, targeting dormant and developing forms of liver and blood stage.

Human malaria is a mosquito-borne disease caused by five species of *Plasmodium* parasites. In 2020, there were approximately 241 million patients affected by malaria worldwide. Of those, 627,000 people died[1]. In sub-Saharan Africa, most cases of malaria and mortality are caused by *Plasmodium falciparum*, while *Plasmodium vivax* is globally ubiquitous and causes persistent infections[2]. *P. vivax* is highly prevalent in the Americas and Southeast Asia, accounting for 75% and 47% of all malaria cases in these regions, respectively[3]. Antimalarials currently available in the clinic include artemisinin-based combination therapies (ACTs), chloroquine phosphate, sulfadoxine/pyrimethamine, mefloquine, primaquine phosphate, halofantrine, and quinine[4]. However, the emergence and persistence of resistance to

[1]Department of Chemistry and Molecular Biology, University of Gothenburg; S-405 30, Gothenburg, Sweden. [2]Center for Global Infectious Disease Research, Seattle Children's Research Institute, Seattle, WA 98109, USA. [3]Seattle Structural Genomics Center for Infectious Disease, Seattle, WA 98109, USA. [4]Molecular Biophysics and Integrated Bioimaging, Berkeley Center for Structural Biology, Advanced Light Source; Berkeley National Laboratory, Berkeley, CA 94720, USA. [5]Mahidol Vivax Research Unit, Faculty of Tropical Medicine, Mahidol University, Bangkok 10400, Thailand. [6]Department of Pediatrics, University of Washington, Seattle, WA 98195, USA. [7]Present address: Department of Chemistry, University of Bergen, Allegaten 41, NO-5007 Bergen, Norway. [8]Present address: School of Biology, Indian Institute of Science Education and Research, Thiruvananthapuram, Kerala 695551, India. [9]These authors contributed equally: Diego Rodríguez-Hernández, Kamalakannan Vijayan. ✉e-mail: Alexis.Kaushansky@seattlechildrens.org; grotli@chem.gu.se

almost all these drugs, and the threat of the continued robust spread of drug tolerance and resistance, necessitate the urgent development of new antimalarials[5,6].

Multiple antimalarials and combination therapies that can be used as single-dose regimens are being developed pre-clinically and clinically. Ambitious efforts to eliminate malaria, such as those by the Bill & Melinda Gates Foundation and the Medicines for Malaria Venture (MMV), include approaches that could target multiple stages of the parasitic life cycle. Yet, the goal of malaria eradication remains unachieved, and several knowledge gaps still need to be addressed in the current pharmacopeia of antimalarial compounds. Antimalarials that are part of several classes are required in order to support a robust malaria eradication campaign. Specifically, candidate drugs targeting the asexual blood stage (target candidate profile [TCP]-1), anti-relapse/hypnozoites (TCP-3), liver schizonts (TCP-4), and transmission-blocking (TCP-5 and TCP-6) are essential for reducing malaria cases worldwide[7,8]. Accordingly, compounds that can achieve multiple objectives would be of particular interest.

Most current drugs do not successfully target the dormant liver stage form[6], called hypnozoites, which are thought to be the source of all relapsing infections[9]. Hypnozoites are commonly observed in *P. vivax* and *Plasmodium ovale* infections. Unlike the developing schizont form of the parasite, hypnozoites remain in the liver for weeks, months, or even years and later reactivate, leading to relapse and symptomatic blood-stage infection. Between 20% and 100% of clinical presentations of *P. vivax* in humans result from hypnozoite relapse, depending on the location and intensity of transmission[10,11]. The current pharmacological options for the elimination of hypnozoites are limited to primaquine and tafenoquine, and the use of these drugs is hampered by their severe toxicity in individuals with glucose-6-phosphate dehydrogenase (G6PD) deficiency[6]. Targeting regulatory proteins within the parasites that are essential for multiple stages of the complex life cycle, including the hypnozoite stage, might be the most robust approach to eliminating malaria.

*N*-myristoyltransferase (NMT) is an essential enzyme in eukaryotes that catalyzes the transfer of myristate from myristoyl-coenzyme A (myrCoA) to the *N*-terminal glycine residue of a nascent polypeptide at the ribosome[12]. Protein myristoylation generally aids in membrane localization as well as protein-protein interactions and stability by increasing the protein's hydrophobicity[13–16]. We and other researchers have previously developed highly active *Plasmodium* NMT inhibitors (NMTis) with moderate selectivity over the human enzyme capable of killing the parasite, some of which have been demonstrated to eliminate *Plasmodium* liver stage and blood-stage parasites[17–23]. Pharmacological inhibition of NMT prevents the completion of blood-stage development of *P. falciparum* partly owing to the inhibition of the inner membrane complex formation[24].

Two potential challenges involved in the development of inhibitors of *Pv*NMT relate to hypnozoite sensitivity and selectivity over human NMTs. Whether similar phenotypic changes occur when NMT is targeted in hypnozoites is unknown. Recent transcriptome studies have shown a remarkable decrease in the number of expressed genes in hypnozoites compared to schizonts[25–27]. NMT is not actively expressed at high levels during this stage. Several potential NMT substrates[20] are expressed, including some essential genes[25–28]. Thus, beyond NMT being essential in *Plasmodium*[24,29], we hypothesize that inhibiting the NMT present in hypnozoites would inhibit several biochemical processes required for the viability of the dormant stage of *P. vivax*.

Although multiple *Plasmodium* NMT inhibitors have been identified, achieving high selectivity against human NMTs remains a major challenge because of the high degree of sequence homology between human and *Plasmodium* peptide binding sites, where all known inhibitors bind. For *Pv*NMT, in particular, the set of residues contacting the inhibitors via their side chains is identical in sequence to the corresponding set of residues in human NMTs[23]. Despite this, early

structure–activity-based studies identified a selective conformational state. A nonselective series and a selective series of inhibitors were shown to bind the side chain of an active site tyrosine in nonrotated and rotated conformations, respectively. A combination of mutagenesis of the tyrosine to alanine and *SI* measurements demonstrated that the rotated state is selective[19]. Inhibitor selectivity was then improved up to twenty-twofold[20] but was difficult to improve further, which has been cited as a reason to deprioritize the targeting of parasitic NMT sites in antimalarial efforts[30]. In fact, because of the high degree of cross-reactivity with *Hs*NMT1, some *Pv*NMT inhibitors were recently repurposed to inhibit *Hs*NMT1 for combating picornaviruses that utilize the host enzyme for *N*-myristoylation[31]. Using a chemical fragment merging scheme, compounds reaching picomolar affinities against *Hs*NMT1 were designed.

In the present study, we applied a hybridization strategy to *Pv*NMT inhibition intending to develop more selective compounds and test their in vitro activity in hypnozoites. The approach is based on a selective scaffold with appended moieties from inhibitors adopting distinct binding modes. We reasoned that such structural perturbations would be compatible with the binding site architecture but would likely introduce molecular stresses that *Pv*NMT might better accommodate due to differences in conformational plasticity with the human NMTs. This approach yielded 11 compounds with selectivity indices (SIs) greater than 100, three exceeding 200, and two exceeding 250. A high-resolution crystal structure provides a molecular basis for the high selectivity, revealing a selective binding site architecture distinct from those of all previously published *Pv*NMT-inhibitor complexes. Finally, selected *Pv*NMTis were evaluated in schizont and hypnozoite infections of liver-stage *P. vivax* and blood-stage *P. falciparum* parasites, showing moderate parasiticidal activity in both stages, validating *Pv*NMT as a multistage target.

## Results

### Design and synthesis of selective *Pv*NMT inhibitors

*Pv*NMT inhibitors DDD85646[20] and IMP-1002[21] (Fig. 1a) were used as the starting point for a structure-based approach to develop more potent and selective *Pv*NMTis. Both NMTis occupy different parts of the *P. vivax* NMT binding pocket, with excellent affinity and potency (Fig. 1b). Additionally, in the presence of IMP-1002, the binding site displays a selective side chain conformation of Tyr211. The molecular hybridization strategy involved appending different head and tail groups to a biaryl core that was also varied chemically. Specifically, head groups were linked to core group A and the tail groups to core group B (Fig. 1c). To target the selective state of Tyr211, the overall topologies of the hybrid compounds reflected that of IMP-1002, with the head and core groups attached to the same sites in the biaryl core.

In the tail region (Fig. 1c, highlighted in blue), the substituents were designed to interact with Ser319[20,22]. In the head region (Fig. 1c, highlighted in brown), the substituents were selected to interact ionically with the carboxylate of the *C*-terminal residue (Leu410), which plays an essential role in myristate transfer and is crucial to the inhibitor's potency against *Plasmodium* NMT[20,21]. Figure 1d shows a particular hybrid compound (**12e**) that was designed and synthesized to dock into the crystal structure of *Pv*NMT with the target hydrogen bonding and ionic interactions formed with Ser319 and Leu410.

Four compound series (**12, 16, 26**, and **30** series) were designed sequentially based on potency and selectivity profiles. All four series tested 1,3,5-trimethylpyrazole, 3,5-dimethylpyrazole, and pyridine moieties in the tail region. Chemical hybridization using the DDD85646 head group, which introduces a 4-atom spacer between *N*4 and the nearest core aryl group (unlike IMP-1002, in which 3 atoms separate the terminal amine and core) first yielded the compound **12** and **16** series. Docking calculations suggested that such compounds would be accommodated in the binding site, although we were motivated to determine the exact binding modes via X-ray crystallography.

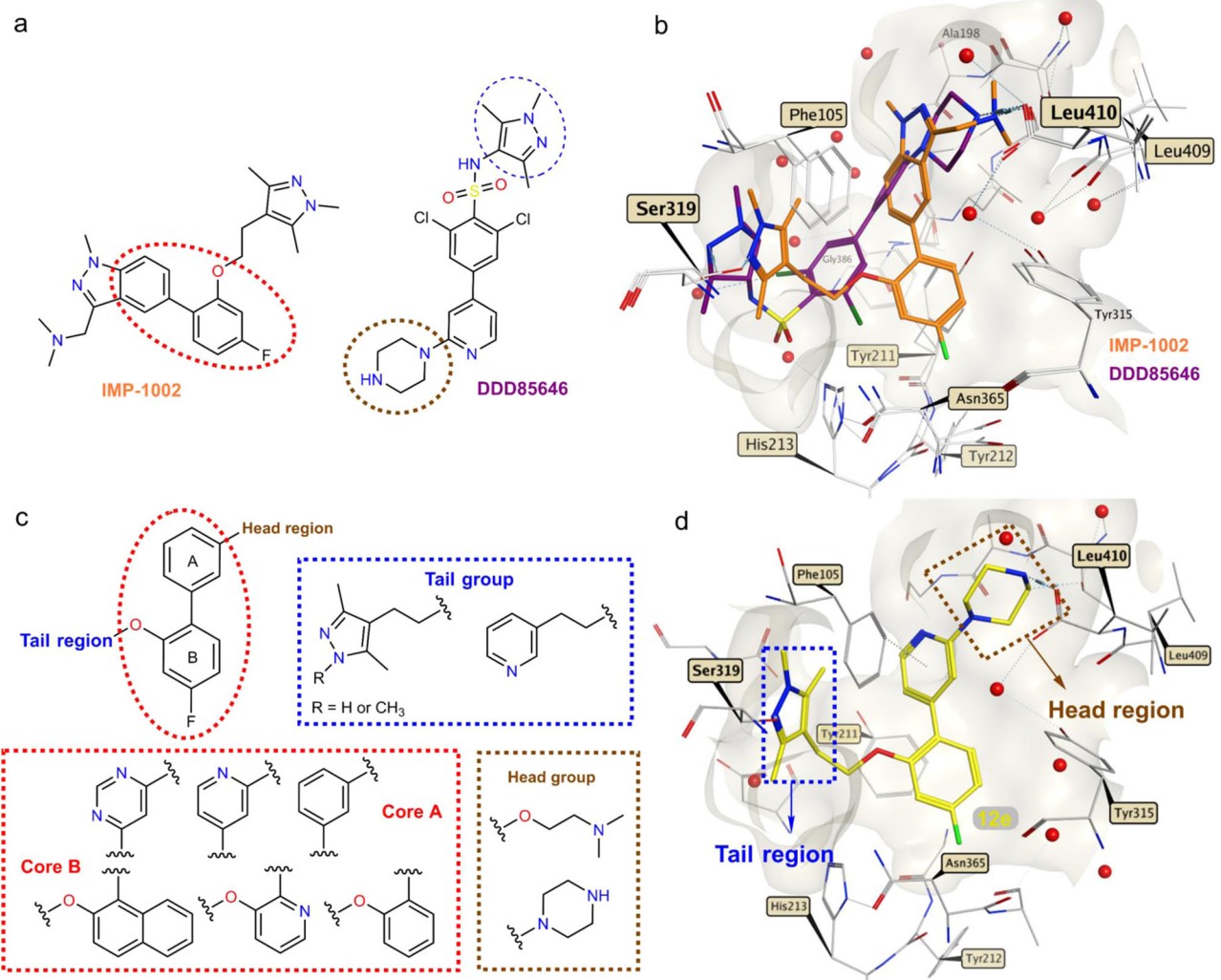

**Fig. 1 | Hybridization approach. a** Chemical structures of IMP-1002[21] and DDD85646[20]. **b** IMP-1002 and DDD85646 bound to *Pv*NMT superimposed (PDB codes 6MB1 and 2YND, respectively). **c** Design of target molecules: cores A and B (red), tail group (blue), and head group (brown). **d** Example of a designed hybrid compound (**12e**) docked into the crystal structure of *Pv*NMT (PDB code 6MB1). Surface representations in front of the binding site are removed for clarity. Interactions involving the atoms of IMP-1002, DDD85646, and compound **12e** are drawn using dashed lines, and red spheres represent water molecules.

Polarity was varied within the compound **12** series at variable positions X, Y and R, and the site where nitrogen is incorporated was found to result in widely varying potencies and selectivities (Supplementary Fig. 2). Polarity was further explored in the compound **16** series through nitrogen insertion at the Z position (Supplementary Fig. 3). The most selective compound of these series (**12b**) possessed a phenyl group as core A and a pyridine group as core B, with a nitrogen atom at the X position (Supplementary Fig. 2). Increasing lipophilicity and bulkiness by appending a phenyl group to core B was tested in the compound **30** series to determine if the gains in selectivity in that region require ring polarity. The effect of the piperazine head group was tested via substitution with dimethylaminoethanol in the compound **26** and **30** series.

**Piperazine-hybrid compounds**

Supplementary Figs. 1–3 summarizes the synthetic routes used to prepare the hybrid compounds (**12a–q** and **16a–f**). These structural fragments were prepared using a three-step synthesis strategy (Supplementary Fig. 1), with a condensation reaction as a key step. The synthesis of hybrid compounds **12a–q** began with the formation of different blocks of aryl ethers (**3–11**) through a sulfonyl transfer[32] or the Mitsunobu reaction[33] (Supplementary Fig. 2). The intermediate blocks

(**3–11**) were obtained from commercially available aryl-hydroxyl or pyridine-hydroxyl with pyrazoles **1** and **2** or with commercially available 3-(2-hydroxyethyl)pyridine. From these aryl-ether blocks, hybrid synthesis was accomplished in two steps (Supplementary Fig. 2). A Suzuki cross-coupling reaction was used to couple cores A and B together (Fig. 1c). This was followed by Boc-deprotection using 4 M HCl in dioxane to furnish compounds **12a–q** in sufficient overall yields (60–91%). NMT activity was indirectly measured through the detection of free CoA by the thiol-reactive probe 7-diethylamino-3-(4′-maleimidylphenyl)-4-methylcoumarin (CPM)[34] (Supplementary Fig. 4).

The enzymatic activities of the hybrid compounds **12a–q** are summarized in Table 1. Five compounds in this series having either a 1,3,5-trimethylpyrazole (**12a**, **12b**, and **12d**) or pyridine (**12g** and **12j**) tail group yielded a selectivity index (SI) of >145, computed as the ratio of IC$_{50}$ values for the *Hs*NMT1 and *Pv*NMT (the value of 145 was the highest SI value obtained against *Pv*NMT from our recent high throughput screening study[18]). Moreover, the three compounds bearing a 1,3,5-trimethylpyrazole tail group had IC$_{50}$ values below 100 nM. Compounds **12a–c**, which increase the core polarity over that of IMP-1002 by incorporating a nitrogen atom at positions X and Y, respectively (Supplementary Fig. 2), display an inverse relationship between potency and selectivity. However, compound **12b**, which

**Table 1 | Biochemical activity of hybrid compounds bearing a piperazine moiety as a head group**

| Compound | $Pv$NMT IC$_{50}$ (nM)[a] | $Hs$NMT IC$_{50}$ (nM)[a] | SI[b] |
|---|---|---|---|
| 12a | 36.8 | 5400 | 146.7 |
| 12b | 80.15 | 20,780 | 259.2 |
| 12c | 9.48 | 599 | 63.1 |
| 12d | 48.5 | 7440 | 153.4 |
| 12e | 15.8 | 124 | 7.8 |
| 12f | 15.9 | 699 | 43.9 |
| 16a | 9390 | – | – |
| 16b | 168 | 27,620 | 164.4 |
| 12g | 104 | 15,900 | 152.8 |
| 12h | 440.6 | >40,000 | 90.9 |
| 12i | 1540 | – | – |
| 12j | 124 | 27,360 | 220.6 |
| 12k | 3120 | – | – |
| 12l | 23.4 | 1710 | 73 |
| 16c | >40,000 | – | – |
| 16d | 3390 | – | – |
| 12m | 12,540 | – | – |
| 12n | 604 | 39,730 | 65.7 |
| 12o | 3116 | – | - |
| 12p | 368 | 23,080 | 62.7 |
| 12q | 73 | 1632 | 22 |
| 16e | 5070 | – | – |
| 16f | 14,033 | – | – |

[a]$Pv$NMT and $Hs$NMT IC$_{50}$ values are shown as mean values of two or more determinations.
[b]Enzyme selectivity calculated as $Hs$NMT IC$_{50}$/$Pv$NMT IC$_{50}$.

**Table 2 | Biochemical activity of hybrid compounds bearing a dimethylamino-ethanol moiety as a head group and hybrid compounds bearing a naphthol moiety in core B of the biaryl scaffold**

| Compound | $Pv$NMT IC$_{50}$ (nM)[a] | $Hs$NMT IC$_{50}$ (nM)[a] | SI[b] |
|---|---|---|---|
| 26a | 266 | >40,000 | 153.8 |
| 26b | 83.2 | 9850 | 118.4 |
| 26c | 29.4 | 4790 | 162.3 |
| 26d | 14,800 | – | – |
| 26e | 12,760 | – | – |
| 26f | 622 | >40,000 | 62.3 |
| 26g | 19,400 | – | – |
| 26h | 1563 | – | – |
| 26i | 242 | 13,860 | 57 |
| 30a | 89 | 24,010 | 269.8 |
| 30b | 19,830 | – | – |
| 30c | 248 | >40,000 | 161 |
| 30d | 7210 | – | – |
| 30e | 2880 | – | – |
| 30f | 25,400 | – | – |

[a]$Pv$NMT and $Hs$NMT IC$_{50}$ values are shown as mean values of two or more determinations.
[b]Enzyme selectivity calculated as $Hs$NMT IC$_{50}$/$Pv$NMT IC$_{50}$.

exhibits the highest selectivity among compounds in this subset and among series **12** compounds more generally (SI = 259), maintains high potency (IC$_{50}$ = 80.2 nM). These results thus demonstrate that increasing selectivity without greatly compromising potency is achievable using the combinatorial design strategy, despite the high degree of active site conservation of $Pv$NMT and $Hs$NMT1.

**Pyrimidine-based hybrid compounds yield mixed results**
Compounds **16a–f** were designed with a pyrimidine moiety as their core A (Fig. 1c) to increase the polarity of the hybrid compounds. The additional nitrogen, relative to related pyridines in the compound **12** series, corresponds to position Z in Supplementary Fig. 2. Structurally, this site is adjacent to the side chain of Tyr211, 4.4–4.5 Å from the hydroxyl group in the $Pv$NMT complex with IMP-1002. To attach the piperazine head to this core, we reacted commercially available 2,4-dichloropyrimidine with Boc-protected-piperazine to generate compound **13** (Supplementary Fig. 3). Subsequent Suzuki coupling with the appropriate boronic acid yielded biaryl intermediates **14** and **15**. The tail components were introduced via the Mitsunobu reaction, followed by Boc-deprotection using 4 M HCl in dioxane, to produce the pyrimidine hybrid compounds (**16a–16f**) in good yields.

In general, the introduction of a pyrimidine group in core A resulted in lower potencies (Table 1). However, compound **16b**, having the same groups in the head and tail regions, was more selective over $Hs$NMT (SI = 164) than **12e** or **12f** (SI = 7.8 and 43.9, respectively). Notably, fluorine substitution at the R site consistently improved potency in this series.

**Dimethylaminoethanol head group shows an increased potency**
With the goal of increasing potency against $Pv$NMT, the piperazine head group was replaced with the 2-(dimethylamino)ethanol moiety.

This increased the cLogP value by approximately 0.3–0.9 log units for the different compounds presented in Table 1. The protonatable nitrogen from the dimethylamino group was expected to form a favorable ionic interaction with the carboxylate group of the $C$-terminal residue. Supplementary Fig. 5 shows the synthetic route used to prepare this series of hybrid compounds (**26a–i**). The synthesis began with a Suzuki coupling reaction using the different blocks of aryl ethers (**3–11**) with the commercially available 3-hydroxyphenylboronic acid to form the biaryl scaffold (**17–25**). Finally, the commercially available 2-(dimethylamino)-ethanol moiety was reacted with a corresponding biaryl framework using the Mitsunobu reaction to produce the desired hybrid compounds (**26a–i**) in good overall yields.

The in vitro IC$_{50}$ values of the hybrid compounds **26a–i** are summarized in Table 2. The three compounds of this series having a 1,3,5-trimethylpyrazole group in the tail region (**26a–26c**) exhibited an SI of >115. Although the compounds with the dimethylaminoethanol group had higher cLogP values than the hybrids that had the piperazine group in the head region (**12f** vs. **26c** or **12d** vs. **26b**), the affinity for $Pv$NMT was similar to that of compounds with the 1,3,5-trimethylpyrazole group in the tail region with IC$_{50}$ values below 100 nM (Tables 1 and 2).

**Naphthol as the core scaffold improves selectivity**
Further synthetic modifications in core B were explored by introducing a polycyclic aromatic hydrocarbon such as naphthol. Through molecular docking of compound **12e** (Fig. 1d), a space was observed in the pocket of the binding site, indicating that it was appropriate to introduce another ring in core B. The introduction of the naphthol group increased lipophilicity, improving potency and selectivity. For these hybrids, different groups of heterocycles were preserved, including the tail moiety and the piperazino-pyridine and 2-(phenoxy)-$N,N$-dimethyl-ethane-1-amine moiety as core A and the head moiety. To synthesize these compounds (**30a–f**), we employed the same reactions presented in Supplementary Figs. 1–3 and 5. Supplementary Fig. 6 displays the synthetic route used to prepare this series of hybrid compounds (**30a–f**) in good overall yields.

The introduction of a naphthol group into core B did not improve the enzymatic activity when the compounds carried

dimethylaminoethanol (**26b** vs. **30b**) in the head region. However, the affinity enhancement imparted by the piperazine group in the head region (**30a** vs. **30b**) was accompanied by an improved selectivity (Table 2). Although compound **30a** had less affinity for *Pv*NMT than compound **12c**, it showed a better degree of selectivity over *Hs*NMT (SI = 269) than did all other hybrid compounds (Tables 1 and 2).

### Compound 12b binds in a selective binding mode

We determined the X-ray co-crystal structure of *Pv*NMT, myrCoA, and **12b** at 1.65 Å resolution (Fig. 2a and Supplementary Table 1). Similar to other *Pv*NMT inhibitors, **12b** binds in a hydrophobic pocket within the peptide binding cleft formed by residues of the *N*- and *C*-terminal domains, including the Ab loop[17,21,23]. Comparison with the structure of inhibitor-free *Pv*NMT shows that conformational changes occur throughout much of the substrate binding sites (Fig. 2a). The inhibitor binding site architecture is unique relative to those of all crystal structures of *Pv*NMT and *Hs*NMT1 thus far characterized, with Tyr211, His213, Phe226, and Leu410 displaying distinct conformations (Fig. 2b).

Compound **12b** aligns closely with that of IMP-1002, reflecting the similarity of their chemical topologies, and its piperazine group binds in a similar location to that of DDD85646 (Fig. 3a, b). However, the *N4* atom of the piperazine, which is spaced by four atoms from the core group, is misaligned with the corresponding amines of the hybrid precursors. As a result, *N4* makes weaker electrostatic interactions with the *C*-terminal carboxylate group (separation distance of 4.0 Å) but is positioned more closely to Thr197, forming a hydrogen bond with the side chain hydroxyl group (Fig. 3c). Additionally, the core group stabilizes a highly rotated conformation of Tyr211, which is also observed in the IMP-1002-bound structure of *Pv*NMT and proposed to represent a selective (*S*) state (Fig. 3d)[18,19,21,35]. This differs from the DDD85646-bound structure, which shows Tyr211 in a nonrotated conformation that corresponds to a nonselective state (*N*)[20] (Fig. 3c). The 1,3,5-trimethylpyrazole tail group of **12b** forms a hydrogen bond with Ser319, similar to that of IMP-1002 and as predicted from the docking calculations performed using **12e**.

The nitrogen atom of the pyridine ring of **12b** forms a hydrogen bond with an additional water molecule accommodated in this region of the binding site relative to that of IMP-1002. This

difference, together with the misalignment of the piperazine with Leu410, is associated with an orientational change in the inhibitor and displacement of Tyr211 (Fig. 3d). The conformational changes permeate both the peptide and myrCoA binding sites. Phe226 is displaced to a position that sterically favors rotamer B of His213, whereas IMP-1002 binding favors rotamer A[21] (Fig. 2b). The conformational differences at Phe226 are associated with larger structural changes at residues 217-247 that narrow the substrate binding cleft at pocket 8 to peptide excluding distances (Fig. 2a and Supplementary Fig. 7).

### NMTis show minimal toxicity in human hepatoma cells

A major concern in developing *Plasmodium* NMT inhibitors is the potential toxicity in host cells due to cross-reactivity with human NMTs, given the structural overlap between the active site of *Plasmodium* and human NMTs[17]. Therefore, we assessed cytotoxicity in the human hepatoma HepG2 cell line. We required a *Pv*NMT to *Hs*NMT SI > 20 for further screening. HepG2 cells were exposed to the compounds at concentrations ranging from 1 to 20 µM for 48 h, and toxicity was assessed via live-dead staining. Staurosporine, a promiscuous kinase inhibitor and known inducer of cell death in HepG2 cells, was used as a positive control; as expected, it reduced the cell viability at 10–20 µM. None of the NMT-targeting compounds induced >30% cell death in the HepG2 cells, even at their highest concentration of 20 µM (Fig. 4a; Source Data file and Table 3).

### NMTis are active against blood-stage parasites

To assess whether any of the *Pv*NMTis developed here could qualify as TCP-1 candidates, we assessed the effect of each compound on blood-stage *P. falciparum* parasites, as there is currently no well-established in vitro platform for screening *P. vivax* blood-stage parasites. Synchronized *P. falciparum* NF54 ring-stage parasites were treated for 72 h with each compound at various concentrations, ranging from 0.625 µM to 10 µM, and parasite DNA replication was measured using a fluorescent DNA binding dye (SYBR Green). The most active compounds against blood stage parasites had the pyridine moiety in core A and the *p*-fluorophenyl group in core B in the biaryl scaffold (**12e, 12l,** and **12i**) (Fig. 4b–d, Source Data file). The relative IC$_{50}$ values for each compound varied from 360 nM to 1.25 µM (Table 3).

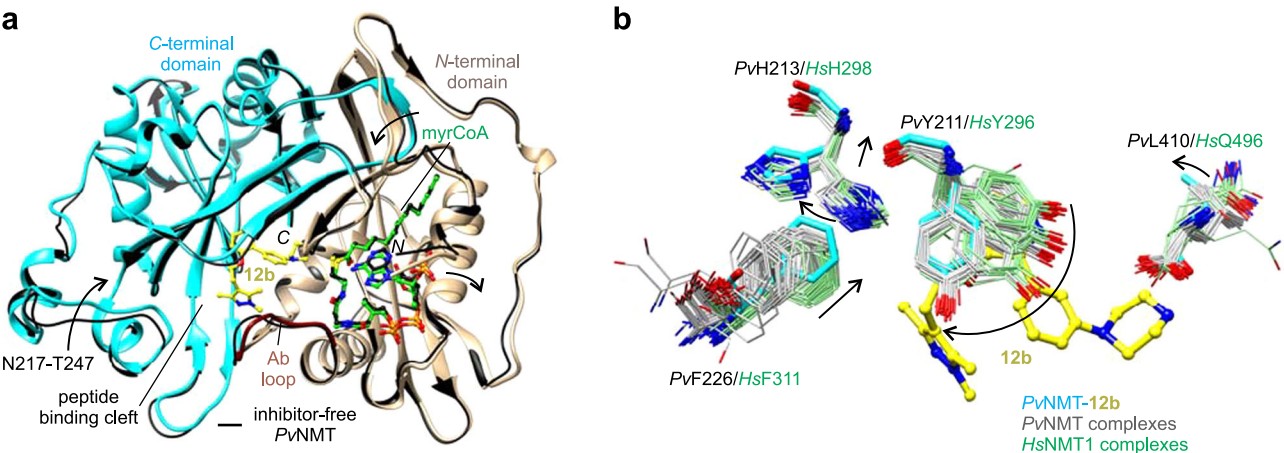

**Fig. 2 | Unique inhibitor binding site architecture exhibited by the *Pv*NMT–12b complex. a** Overall architecture. **12b** binds in the substrate peptide binding cleft at the interface of the NMT *N*- and *C*-terminal domains (tan, residues 1–204, truncated at residue 27, and cyan, residues 205–410, respectively), with the Ab loop (dark red, residues 95–102) in the closed conformation. Comparison with the inhibitor-free state [thin black ribbons, PDB entry 4B10[19]] shows significant conformational changes in the substrate binding sites (indicated schematically with arrows). **b** Distinct conformational changes in selected residues, namely *Pv*Tyr211, *Pv*His213,

*Pv*Phe226, and *Pv*Leu410. Carbon atoms of the following crystal structures of *Pv*NMT and *Hs*NMT1, which were superimposed onto the structure of *Pv*NMT-**12b** using Chimera[51], are colored gray and light green, respectively: 2YNC/D/E[20], 3IU1, 3IU2, 3IWE, 3JTK, 4A95[23], 4B10/1/2/3/4[19], 4BBH[53], 4C2Y/Z[54], 4C68[55], 4CAE/F[22], 4UFV/W/X[56], 5G1Z/22[57], 5MU6/O48/O4V/O6H/O6J[31], 5NPQ, 5O9S/T/U/V[58], 5UUT[59], 5V0W, 5V0X, 6B1L, 6FZ2/3/5[60], 6MB1[21], 6NXG[18], 6PAV[61], 6EHJ/QRM/SJZ/SK2[62], 6TW5, 6TW6, and 7RK3.

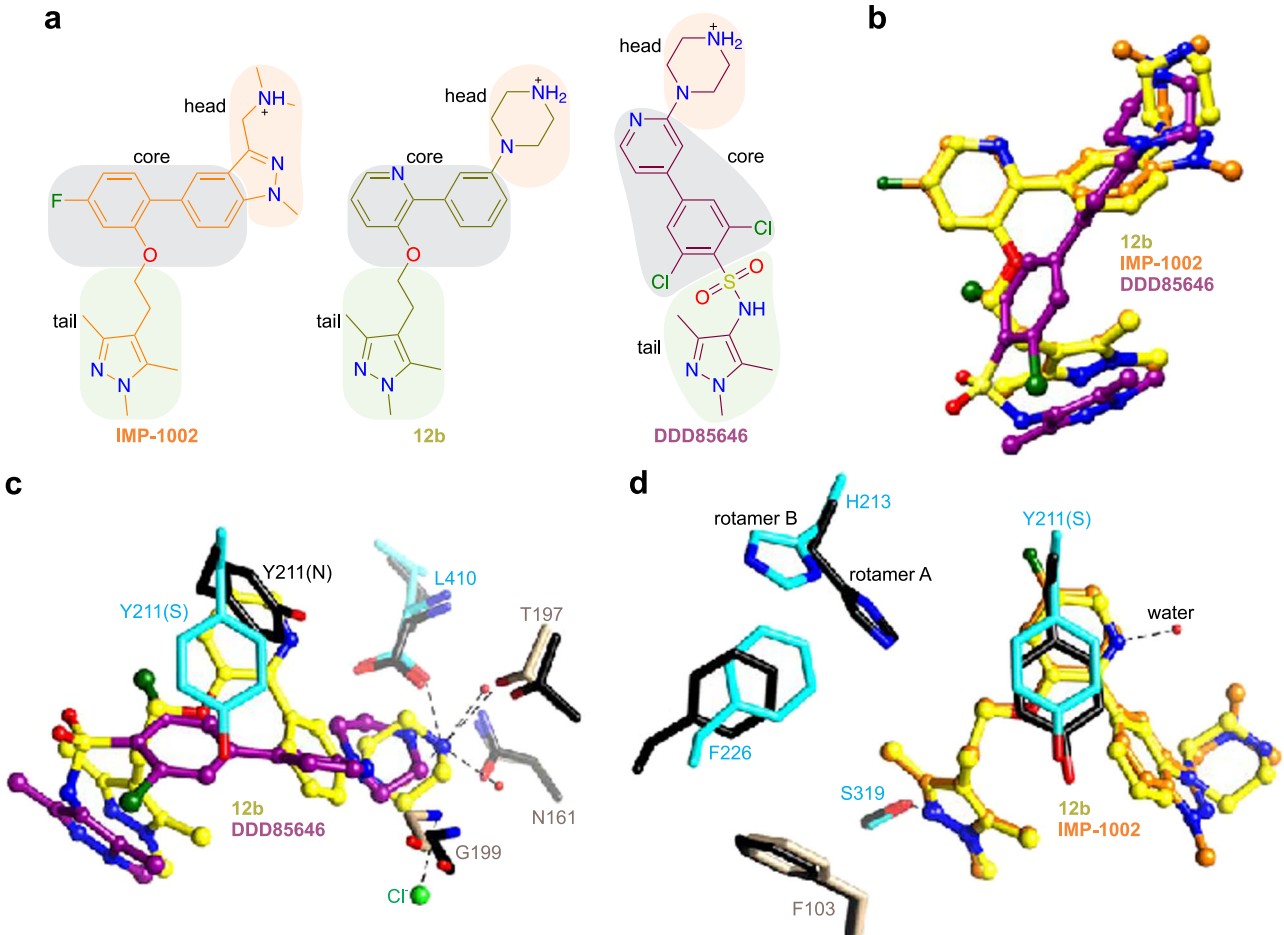

**Fig. 3 | Hybridization and the selective state stabilized by 12b. a** Compound **12b** contains the head group of DDD85646, the tail group of IMP-1002, and has a similar topology as IMP-1002. **b** Compound **12b** adopts a similar binding site pose as IMP-1002 but displays an orientational change in its biaryl core. The piperazine group occupies a similar binding site location as that of DDD85646, but the *N4* atoms are misaligned, reflecting the similar pose to that of IMP-1002 but difference in spacer lengths. **c** Misalignment of the piperazine group with that of DDD85646 results in different interactions with nearby polar sites of *Pv*NMT via *N4*. **d** Both **12b** and IMP-1002 stabilize the *S* state of Tyr211, but the pyridine moiety of **12b** incorporates an additional water molecule, which accompanies the ligand orientational change. Additional conformational differences occur in Tyr211, His213, and Phe226. In panels **c**, **d**, aligned residues of IMP-1002- and DDD85646-bound *Pv*NMT are colored black [PDB entries 6MB1[21] and 2YND[20], respectively]. Hydrogen bonding and salt bridge interactions involving **12b** are depicted using dashed lines.

## NMTis are active against *P. vivax* liver-stage schizonts

We next evaluated the candidate compounds against liver-stage *P. vivax* (Fig. 5). *P. vivax* sporozoites were obtained from three independent patient isolates and cultured in primary human hepatocytes from a single donor lot using a 384-well microculture system as described previously[36]. Cells were infected and treated with NMTis at 20 μM beginning day 5 post-infection until day 8 when the infection rates were quantified by fluorescent microscopy. We defined schizonts as any liver-stage parasites that exhibited circumferential *P. vivax* Upregulated in Infectious Sporozoites−4 (*Pv*UIS4) staining, had multiple nuclear masses, and were ≥10 μm in diameter (Fig. 4a).

We observed robust schizont infections in each of the three independent patient isolates (Fig. 5b, Source Data file) and found that all the tested compounds reduced the parasite load by at least 90% when administered at 20 μM (Fig. 5b and Table 3). Accordingly, we applied dose–response curves of all the tested NMTs on parasite isolates 2 and 3 over concentrations up to 20 μM (Fig. 5c, d, Source Data file). We observed similar kinetics for all inhibitors, with a gradual decrease in parasite load upon increasing the drug concentration. The IC$_{50}$ values were calculated for each compound and ranged from 2.2 to 6 μM (Table 3). Nuclei were counted in each well using DAPI staining as a measure of toxicity (Supplementary Fig. 86, Supplementary Source data). We used the phosphatidylinositol 4-kinase inhibitor (PI4Ki),

MMV390048, as a positive control for eliminating schizonts[37,38]. As expected, the PI4Ki dramatically reduced the number of schizonts per well. The most active compounds in this stage (**12b** and **12g**) shared the same scaffold and the piperazine moiety in the tail region; both compounds showed an SI of >150 in the biochemical assay (Table 1).

## NMTis are active against *P. vivax* liver stage hypnozoites

Finally, we evaluated the NMT-binding compounds against non-developing *P. vivax* hypnozoites. Hypnozoites were defined as parasites having a single nucleus, diameter of <10 μM, and exhibiting prominent staining for *Pv*UIS4 at a point within the parasite periphery (termed the 'prominence')[39] (Fig. 6A). Of the three parasite isolates, we used to infect primary human hepatocytes; two isolates exhibited robust numbers of *P. vivax* hypnozoites as defined by these criteria (Fig. 6B, Source Data file). We, therefore, evaluated the impact of all tested compounds on *P. vivax* hypnozoites using these two isolates. All tested compounds except **26i** and **30c** exhibited >90% inhibition of hypnozoites at 20 μM (Fig. 6C, Source Data file). Compound **30c** exhibited >75% inhibition at 20 μM. The IC$_{50}$ values were calculated for each compound and ranged from 1.2 to 12 μM (Fig. 6D and Table 3). Six compounds showed IC$_{50}$ values < 2.1 μM in this stage, two of which (**12b** and **30a**) had an SI of >250 in the enzymatic assay, both having the same group in the tail and head regions (Tables 1 and 2).

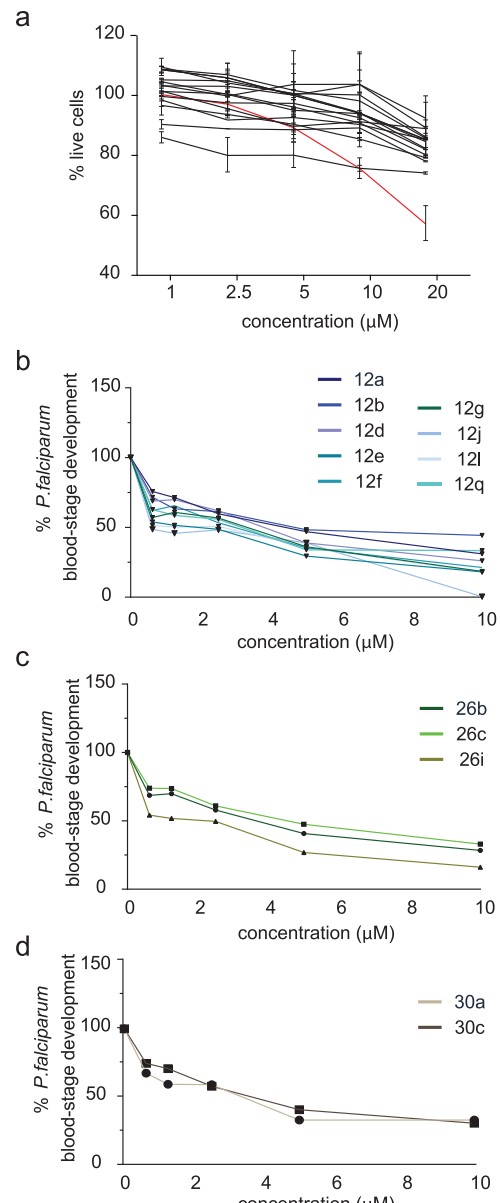

**Fig. 4 | NMT inhibitors show minimal toxicity in human hepatoma cells and reduce *Plasmodium falciparum* blood-stage development. a** Percentage of live HepG2 cells after 48-hour treatment with each NMT inhibitor (shown in black) compared with that of controls treated with diluent. Data are presented as mean values ± SD. The staurosporine-positive control is shown in red. **b–d** Percentage change in *P. falciparum* blood-stage development as measured by parasite DNA level fluorescence assay after 72-hour treatment with NMT inhibitors from each of the three prioritized compound families. Data are normalized to untreated controls set at 100%. Source data are provided as a Source Data file.

**Table 3 | IC$_{50}$ values of NMT inhibitor compounds for cytotoxicity and effect on *Plasmodium* blood and liver stage infection**

| Compound | Cytotoxicity | Blood stage | Liver stage | |
|---|---|---|---|---|
| | HepG2 IC$_{50}$ (µM) | *P. falciparum* NF54 IC$_{50}$ (nM) | *P. vivax* schizonts IC$_{50}$ (µM) | *P. vivax* hypno-zoites IC$_{50}$ (µM) |
| 12a | >20 | 810 | 3–3.8 | 5.7 |
| 12b | >20 | 470 | 2.3–4.6 | 1.7 |
| 12d | >20 | 800 | 3.3–4.3 | 4.7 |
| 12e | >20 | 360 | 3.4–3.7 | 1.2 |
| 12f | >20 | 610 | 3.3–4.3 | 1.5 |
| 12g | >20 | 600 | 2.2–3.1 | 1.2 |
| 12j | >20 | 1250 | 4.9–6.0 | 12 |
| 12l | >20 | 370 | 2.9–4.7 | 2.1 |
| 12q | >20 | 380 | 2.2–3.5 | 4.4 |
| 26b | >20 | 710 | 3.1–4.1 | 4.6 |
| 26c | >20 | 820 | 2.8–4.2 | 4.2 |
| 26i | >20 | 400 | 3.3–5.2 | 11 |
| 30a | >20 | 440 | 2.9–4.3 | 2.1 |
| 30c | >20 | 680 | 2.7–4.9 | 3.8 |

against multiple life cycle stages of *Plasmodium*, and across evolved field isolates.

## Discussion

The development of potent anti-malarial compounds that target the complete life cycle of *Plasmodium* is needed for eradication. As most of the currently licensed antimalarials target only the erythrocytic stage, expanding our antimalarial arsenal is crucial. Drugs that can effectively target the liver stage have been a particular challenge to develop. The liver stage is a clinically silent and obligatory developmental phase that occurs before parasites can infect erythrocytes and cause malaria symptoms. Targeting the hepatic stage is therefore highly desirable in the context of malaria eradication, not only because its asymptomatic nature makes it ideally suited for prophylactic intervention but also because the liver can serve as a reservoir for *P. vivax* hypnozoites, the dormant parasite forms that cause relapses long after the initial blood infection has been treated.

The enzyme NMT is expressed throughout the *Plasmodium* life cycle and is a promising target for antimalarial drug development. Schlott and colleagues showed that NMT inhibition disrupts at least 3 vital pathways of the parasite lifecycle: early schizont development, merozoite formation, and merozoite egress[24]. Despite this promise, the enzyme has been deprioritized as an anti-malarial target due to challenges in identifying parasite-selective inhibitors and a slow mechanism of action[30]. The impact of NMT inhibition on hypnozoites has not been fully investigated. Here we demonstrate that NMT inhibitors reduce the growth of *P. vivax* schizonts and hypnozoites derived from clinical isolates, making *Pv*NMT a potential target for eliminating dormant forms of the parasite. Combining NMT inhibitors with fast-killing blood stage compounds is an attractive strategy for complete parasite elimination[30].

To develop these selective inhibitors, we utilized a hybrid approach to combine previous inhibitor designs to maximize the affinity and selectivity of the parasite over the human enzyme. The combinatorial designs included three tail group moieties (1,3,5-tri-methylpyrazole, 3,5-dimethylpyrazole, and pyridine) and two head group moieties (piperazine and dimethylaminoethanol), and the core region was varied extensively at four unique positions (Figs. 1c and 7a). The use of the piperazine head group (hybridization of IMP-1002 and DDD85646) was unique because of the longer, 4-atom spacer

Interestingly, the efficacy of NMTis on schizont (Fig. 5) and hypnozoite (Fig. 6) *P. vivax* liver stage forms, as well as efficacy against *P. falciparum* asexual blood stages (Fig. 4), was strongly correlated (Fig. 6E). NMTs also exhibited consistent effects across *P. vivax* parasite isolates. Multiple comparison tests for each compound revealed no significant difference in the effect of NMTis on schizonts between isolates, except for compound **12q** (Supplementary Table 2). A significant difference in the effect on hypnozoites between isolates was observed for compound **26i** only. Notably, compound **26i** was the only inhibitor that showed a significantly different effect on the two parasite forms, suggesting that *Pv*NMT targeting compounds might form the basis for a drug that is effective

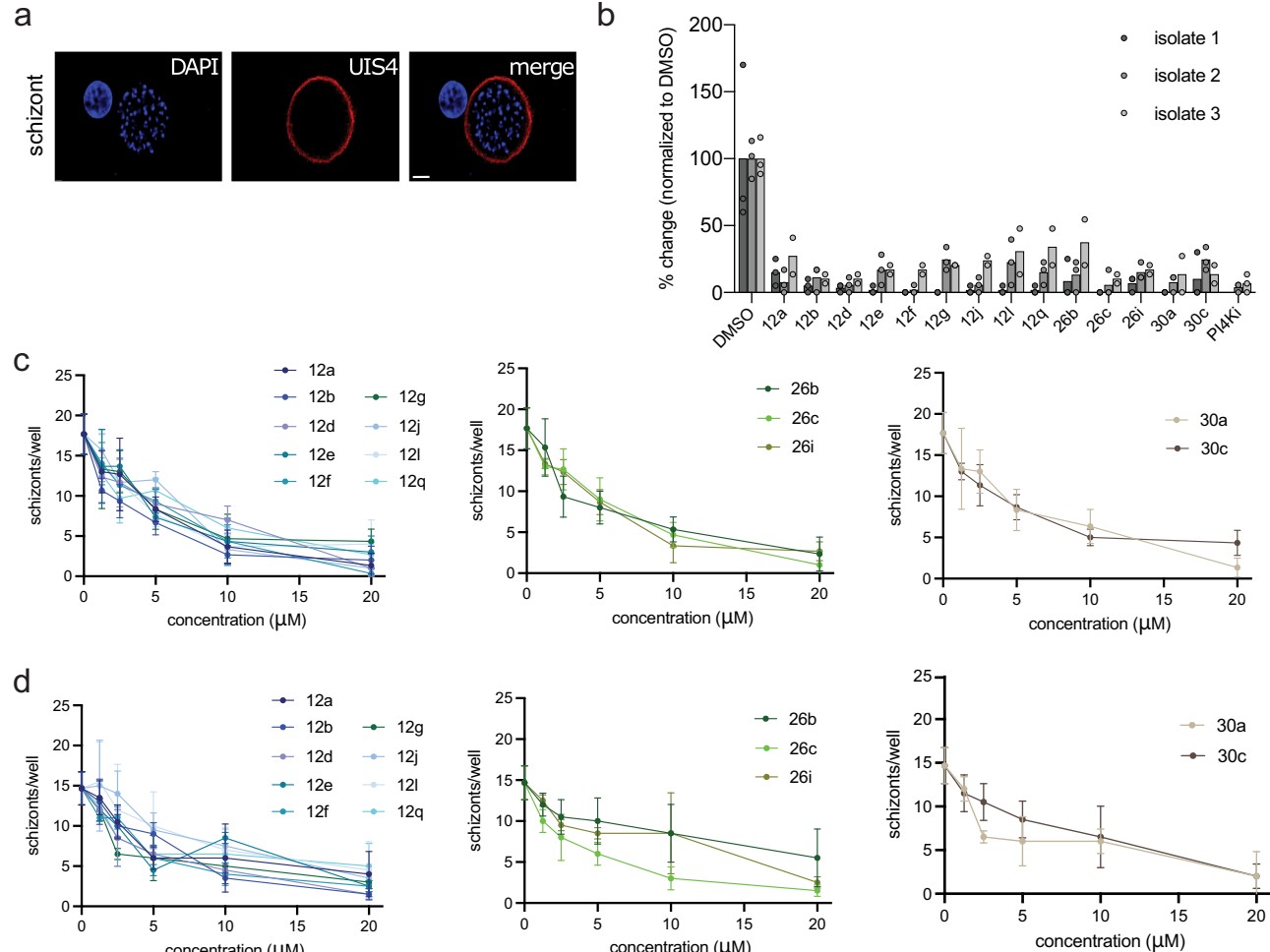

**Fig. 5 | NMT inhibitors reduce *Plasmodium vivax* schizont infection in vitro.**
**a** Representative image of a schizont parasite at 8 dpi. Scale bar: 10 μm. **b** Effect of NMT inhibitor compounds on schizont infection levels at 8 dpi in three independent parasite isolates. All compounds were used at 20 μM, with DMSO as a control. Data from each isolate are normalized to the average of their DMSO control and shown as the mean value with three (isolates 1–2) or two (isolate 3) technical replicates overlaid. **c**, **d** Dose–response curves for each compound, grouped by family for isolates 2 and 3; *n* = 3. Data are shown as mean ± SD. Source data are provided as a Source Data file.

introduced between the terminal amine and core regions. Notably, docking calculations suggested that such designs can be accommodated by the binding site (Fig. 1d), and remarkably, six of these compounds analyzed in vitro displayed $IC_{50}$ values against *Pv*NMT of less than 90 nM and reached selectivity indices relative to human NMT1 above 115 (Fig. 7b). Moreover, six compounds displayed high efficiencies against both blood and liver stages of *P. vivax* (Fig. 7b). Compounds **12b** and **30a** belong to both groups and thus emerge as the most favorable lead inhibitors (Fig. 7b).

The high-resolution crystal structure of *Pv*NMT bound to **12b** demonstrates that a hybrid compound can bind within the active site. The high selectivity is likely attributed to the distinct way the core and head groups interface with the adjacent waters and polar sites near the carboxy terminus, which accompanies conformational changes that propagate throughout the peptide and myrCoA binding sites. The selectivity of compound **30a**, the most selective inhibitor of the presently developed series, is also expected to leverage the adjacent water channel via the insertion of its naphthalene group. These two compounds, and close derivatives, should also be considered for selectivity analysis against *P. falciparum* NMT. There are presently no crystal structures of *Pf*NMT available in the Protein Data Bank, but a comparison of amino acid sequences (UniProt accession codes Q8ILW6 and A5K1A2) shows only one amino acid replacement in the **12b**

binding site—a phenylalanine substitution at the *Pv*NMT Tyr334 site. A key question is whether the same mechanisms leading to the high selectivities of **12b** and **30a** will act in the peptide binding cleft of *Pf*NMT and if the phenylalanine substitution can have an additive effect.

Our study not only provides insights into how selective *Pv*NMT inhibitors can be developed but also underscores the potential value of NMTis as antimalarial drugs. While several *P. vivax* targets have been identified, and strategies to interfere with the non-developing *P. vivax* hypnozoite form remain a major gap in drug development. Therefore, it is particularly impactful that the compounds we describe here target both developing and dormant liver forms of *P. vivax*. In conclusion, we have developed NMTis that show promise as multi-stage targeting antimalarials, and our work suggests *P. vivax* NMT may be a target for anti-relapse therapy.

## Methods
### Ethical statement
Plasmodium vivax infected blood was collected from patients attending malaria clinics in Tak and Yala provinces, Thailand, under the approved protocol by the Ethics Committee of the Faculty of Tropical Medicine, Mahidol University (MUTM 2018-016-03). Written Informed consent was obtained from each patient before sample collection.

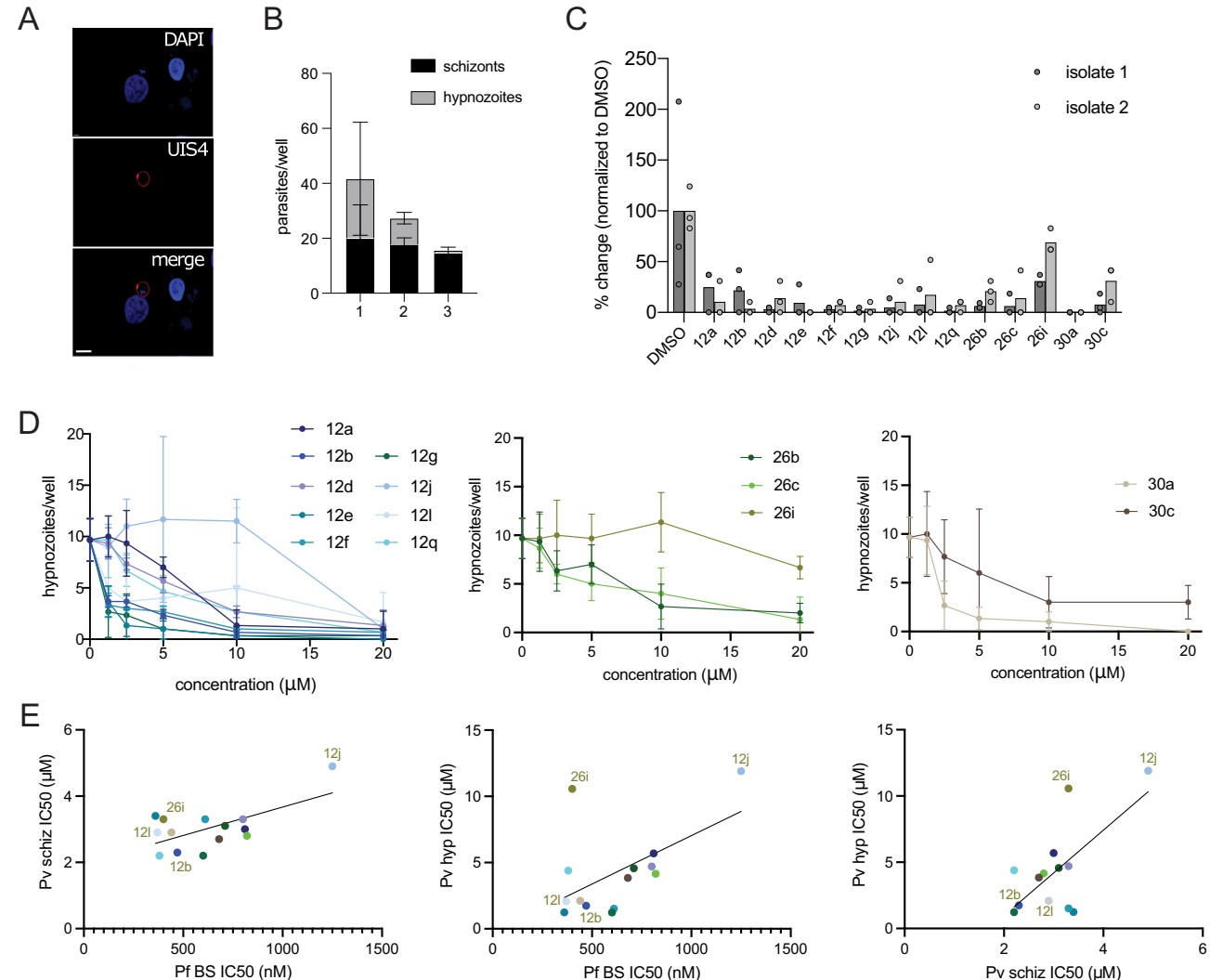

**Fig. 6 | NMT inhibitors reduce *P. vivax* hypnozoite forms in vitro.**
**A** Representative image of hypnozoite form 8 dpi. Scale bar: 10 μm. **B** Schizont and hypnozoite infection levels for each isolate. Data are presented as mean values ± SD. **C** Effect of NMT inhibitor compounds on hypnozoite infection levels 8 dpi for 2 independent parasite isolates. All compounds were administered at 20 μM with DMSO as a control. Data from each isolate are normalized to the average of their DMSO control and are shown as mean values overlaid with 3 (isolate 1) or 2 (isolate 2) technical replicates. **D** Dose–response curves for each compound, grouped by family, for isolate 2. Data are shown as mean values ± SD; $n = 3$. **E** Correlations between $IC_{50}$ against *P. falciparum* blood-stage (Pf BS), *P. vivax* schizont (Pv schiz), and *P. vivax* hypnozoite (Pv hyp) infection plotted against each other for each NMTi. Source data are provided as a Source Data file.

## Synthetic procedure and characterization of compounds

All the hybrid compounds were synthesized as described in Supplementary Methods, and the NMR spectra for the final compounds (**12a–12q, 16a–16f, 26a–26i**, and **30a–30f**) are illustrated in Supplementary Figs. 9–84.

## Docking of compound 12e into the binding pocket of *Pv*NMT

The hybrid compound **12e** was rendered in the form of 2D images using the ChemDraw Professional (Version 19.1.1.21) software package, converted to SDF format, and then prepared for docking using the Molecular Operating Environment (MOE 2019.01) software package. After loading the SDF files, it was processed as follows: the compound was energy-minimized, and partial charges were added (Amber10 forcefield) using QuickPrep. To prepare the receptor protein, the *Pv*NMT PDB file (PDB: 6MB1) was loaded into MOE and processed using QuickPrep. The docking simulation was set up by setting the receptor to "receptor+solvent." The SDF file containing the processed ligands to be docked was loaded. Ligand placement and refinement were performed using the Alpha PMI and rigid receptor methods, with 30 and 3 poses, respectively.

## Cloning, expression, and purification of NMT enzymes

Cloning, expression, and purification were conducted as part of the Seattle Structural Genomics Center for Infectious Disease (SSGCID)[40,41] following protocols described previously[18,21,42,43]. A region of the *Pv*NMT gene encoding residues 27–410 with an *N*-terminus 6xHis sequence and PreScission cleavage site was codon optimized for expression in *E. coli* and cloned into a pET11a expression vector (GenScript) as described previously[18,21]. The *N*-terminal sequence is MGSSHHHHHHSAALEVLFQ/GP-ORF, where cleavage occurs between the glutamine and glycine residues. Plasmid DNA was transformed into chemically competent *E. coli* Rosetta 2 (DE3) pRARE cells. Cells were expression tested, and 4–12 liters of culture were grown using auto-induction media[44] in the LEX bioreactor for 18-22 h at 18 °C. The expression clone was assigned the SSGCID target identifier Plvi-B.18219.a.FR2.GE44010 and is available at https://www.ssgcid.org/available-materials/ssgcid-proteins/. Protein was purified following a 5-step procedure as previously described[18,21] consisting of Ni²⁺-affinity chromatography (IMAC), cleavage of the 6xHis-tag and pass over a second IMAC column to remove cleaved tag and protease. The eluted protein was purified further using an anion exchange HiTRAP Q HP

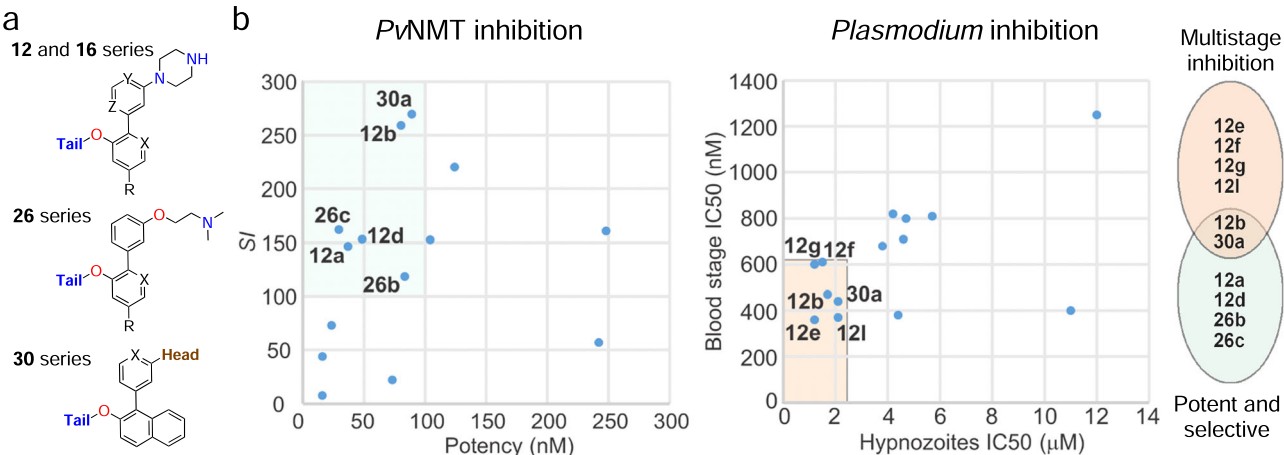

**Fig. 7 | Hybrid *Pv*NMT inhibitors reach high potencies and selectivities. a** Four series of compounds synthesized via chemical hybridization. **b** In vitro inhibitor selectivity indices (*SI*s) and potencies against *Pv*NMT (left graph), and in vitro inhibition concentrations of blood and liver stages of *Plasmodium* (right graph). Inhibitors displaying favorable inhibition profiles are labeled. Venn diagram highlights **12b** and **30a** as the most promising lead inhibitors.

5 mL column. Peak fractions were concentrated to 5 mL and applied to a Superdex 75 10/300 column. The final buffer was composed of 0.3 M NaCl, 20 mM HEPES, 5% (v/v) glycerol, 1 mM TCEP, and pH 7.0. Fractions were analyzed on an SDS-PAGE gel, and fractions containing the target protein were concentrated and flash frozen, and stored at −80 °C until further use.

**Crystallization and structure determination of *P. vivax* NMT**
Purified *Pv*NMT (residues 27–410) concentrated to 8 mg/mL was incubated with 1 mM myrCoA (MedCoA101 LLC.) and compound **12b** for 20 min at room temperature and set up in 96-well sitting drop crystallization screens JCSG + HT96 (Molecular Dimensions) and Morpheus HT96 (Molecular Dimensions). Crystals formed within 2 weeks in JCSG+ condition A1 composed of 0.2 M lithium sulfate, 0.1 M sodium acetate, and 50% PEG 400, pH 4.5. Crystals were harvested directly, and flash frozen in liquid nitrogen without cryo-protectant exchange. Frozen crystals were shipped to the Advanced Light Source (ALS), Berkeley National Laboratory as part of the Collaborative Crystallography program of ALS-ENABLE. Data were collected at beam line 8.2.2. Mounted crystals were maintained at 100 K and irradiated with X-rays having a wavelength of 1.00000 Å. Additional details are provided in Supplementary Table 1. Data were indexed and integrated with HKL2000[45] and scaled with XSCALE[46]. The structure was solved with Phaser[47] using PDB 6NXG as a search model. The model was refined with iterative rounds of refinement with Phenix[48] and manual model building in Coot[49]. The quality of the structure was checked with Molprobity[50]: 97.13% of the backbone torsional angles were determined to be in the favored regions of Ramachandran space, with the remaining angles in the allowed regions; 97.93% of the modeled residues displaying favored rotamers. The structure of **12b** was validated through the aid of a 2*Fo−Fc* composite omit map computed using Phenix[48] (Supplementary Fig. 8). Molecular graphics of the crystal structure were produced using Chimera[51].

**NMT activity assay**
To measure the activity of the purified *Pv*NMT, an assay was adapted from Goncalves et al.[34]. The assay buffer was prepared in a 4× stock solution consisting of 9.2 mM potassium phosphate, 69.7 mM sodium phosphate, 2 mM EDTA and 10% TritonX-100 at pH 7.0. Working stock solutions were made fresh, adding DMSO for final concentrations of either 1% or 5% DMSO. The *Pv*NMT enzyme was diluted in an assay buffer containing 1% DMSO for a final concentration of 25 nM. Ten microlitres of the test compound or 10% (v/v) DMSO/water were dispensed into a 96-well plate (Greiner Bio-One) and 50 μL of the enzyme

(in assay buffer containing 1% DMSO) was added for a final concentration of 25 nM per well. The plate was incubated for 30 min at room temperature. The enzymatic reaction was initiated by adding 50 μL of reaction substrate containing 10 μM myrCoA and *Pf*ARF, as well as 8 μM CPM. Fluorescent readings were taken on a Spectra M2 plate reader (Molecular Devices) with excitation at 385 nm and emission at 485 nm. Fluorescent intensity was measured continuously in one-minute intervals for 45 min. Background fluorescence and noise were determined by replacing each constituent of the reaction individually with an assay buffer containing 1% DMSO, and values were deducted from experimental samples. The enzymatic reactions were set up with varying pHs (6.0–8.5) of the assay buffer to determine optimal reaction conditions with minimal background and off-target reactions. In lieu of compound, 10 μL of 10% DMSO/H$_2$O was added to each well. The fluorescence signal was obtained continuously for 45 min. A linear reaction rate was observed during the first 30 min and used to determine all values. The synthetic peptide (*Pf*ARF) Gly-Leu-Tyr-Val-Ser-Arg-Leu-Phe-Asn-Arg-Leu-Phe-Gln-Lys-Lys-NH$_2$ was purchased from Innopep (San Diego, California). 7-Diethylamino-3-(4′-Maleimidylphenyl)-4-Methylcoumarin (CPM) was purchased from Thermo Scientific Life Technologies (Grand Island, New York), and the co-factor myrCoA was purchased from Med Chem 101 LLC (Plymouth Meeting, Pennsylvania). IC$_{50}$ calculations were calculated using Prism (GraphPad Software, Inc.).

**HepG2 cytotoxicity assay**
HepG2 (HB-8065) cells were obtained from ATCC. The viability of the HepG2 cell line following exposure to the compounds was determined by a Live/Dead cell assay kit (Invitrogen). Briefly, cells were added to a 96-well plate at a concentration of 5000 cells per well, excluding the exterior wells. Cells were exposed to the compounds at different concentrations ranging from 1 to 20 μM. Cells were washed every 24 h and supplemented with fresh doses of compounds. At 72 h post-treatment, 1 μM calcein-AM and 1 μM ethidium homodimer were added to the wells and incubated for 10 min at 37 °C. After washing with PBS, the cells were visualized with fluorescent microscopy (Keyence BZ-X700).

**P. falciparum blood stage assay**
Sorbitol-synchronized *P. falciparum* NF54 ring-stage parasites were cultured at a parasitemia of 0.5% and hematocrit of 1.5% in a 96-well microtiter plate. Parasites were treated with each NMT inhibitor compound at 0.625, 1.25, 2.5, 5, or 10 μM for 72 h. Direct lysis of the blood cells to release the parasite DNA was performed by adding

100 μl of LBS buffer containing SYBR green I DNA binding dye to each well. Plates were incubated at −20 °C overnight for complete lysis, and fluorescence was measured at 485 nm (excitation) and 528 nm (emission). ICEstimator regression analysis (http://www.antimalarial-icestimator.net/) was used to obtain relative $IC_{50}$ values for each compound.

## Generation of *P. vivax* sporozoites
*Plasmodium vivax-infected* blood was collected from patients attending malaria clinics in Tak and Yala provinces, Thailand, under the approved protocol by the Ethics Committee of the Faculty of Tropical Medicine, Mahidol University (MUTM 2018-016-03). Written Informed consent was obtained from each patient before sample collection. The infected blood was washed once with RPMI1640 incomplete medium before being resuspended with AB serum to a final 50% hematocrit and fed to a female *Anopheles dirus* through membrane feeding. The engorged mosquitoes were maintained in a 10% sugar solution until used. The infected mosquitoes at day 14–21 post-feeding were used for harvesting sporozoites.

## *P. vivax* liver stage assay
*Plasmodium vivax* liver stage assays were performed as described previously[36]. Cryopreserved primary human hepatocytes (Cat No. F00995-P) and hepatocyte culture medium (HCM) (InVitroGroTM CP Medium) were obtained from Bioreclamation IVT, Inc., (Baltimore, MD, USA). Briefly, we seeded 384-well plates with primary hepatocytes from a single donor lot at a density of 25,000 cells per well. We then infected the hepatocytes with 14,000 freshly hand-dissected sporozoites per well. Cells were exposed to the compounds at indicated concentrations from day 5 post-infection. We used Pi4K inhibitor MMV390048, at 20 μM, as a positive control to eliminate *P. vivax* liver schizonts. We then proceeded to feed cultures every day, replacing the compounds until day 8 post-infection. Cells were fixed with 4% PFA, permeabilized with 1% Triton X-100, blocked with 2% BSA, and stained with DAPI and *Pv*UIS4 antibodies to detect schizonts and hypnozoites forms by confocal microscopy. For antibody dilutions, PvUIS4 (kind gift from Dr. Noah Sather, Seattle Children's Research Institute) was diluted 1:250 and the secondary antibody was anti-mouse AlexaFluor-594 (Cat # A-11005; RRID: AB_2534073) diluted 1:1000. DAPI was diluted 1:2000. The images were processed, analyzed and outlier cells were removed using IMARIS (Bitplane Inc.) image analysis software.

## Statistical analysis
A two-way ANOVA with Dunnett's multiple comparisons tests (unpaired) was used to compare the effect of each NMTi on schizonts compared to DMSO controls and to compare the effect of each NMTi on schizonts between parasite isolates. Sidak's multiple comparisons test (paired) was used to compare the effect of each NMTi on hypnozoites and schizonts within an isolate.

## Reporting summary
Further information on research design is available in the Nature Portfolio Reporting Summary linked to this article.

## Data availability
The atomic coordinates and structure factor amplitudes of the crystal structure of *Pv*NMT bound to myrCoA and **12b** are available at the Protein Data Bank, accession code 8FBQ. The associated raw X-ray diffraction images have been deposited in the Integrated Resource for Reproducibility in Macromolecular Crystallography under accession code 8FBQ at https://proteindiffraction.org[52]. Protein expression plasmids of *Pv*NMT, *Hs*NMT1, and *Hs*NMT2 are available at https://www.ssgcid.org/available-materials/ssgcid-proteins/ through a material transfer agreement (MTA). Source data are provided in this paper. The UniProt accession codes for *Pv*NMT and *Pf*NMT are A5K1A2 and Q8ILW6, respectively. The accession codes of all other PDB coordinate files referenced in this study are: 2YNC, 2YND, 2YNE, 3IU1, 3IU2, 3IWE, 3JTK, 4A95, 4B10, 4B11, 4B12, 4B13, 4B14, 4BBH, 4C2Y, 4C2Z, 4C68, 4CAE, 4CAF, 4UFV, 4UFW, 4UFX, 5G1Z, 5G22, 5MU6, 5O48, 5O4V, 5O6H, 5O6J, 5NPQ, 5O9S, 5O9T, 5O9U, 5O9V, 5UUT, 5V0W, 5V0X, 6B1L, 6FZ2, 6FZ3, 6FZ5, 6MB1, 6NXG, 6PAV, 6EHJ, 6QRM, 6SJZ, 6SK2, 6TW5, 6TW6, and 7RK3. Source data are provided in this paper.

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

## Acknowledgements

We thank Dr. Noah Sather for the *P. vivax* UIS4 antibody and Dr. Dennis Kyle and Dr. Steven Maher for the PI4K inhibitor. This research used the Advanced Light Source resources, a DOE Office of Science User Facility under contract No. DE-AC02-05CH11231. The ALS-ENABLE beamlines are supported by the National Institutes of Health, National Institute of General Medical Sciences, grant P30 GM124169-01. This project has been funded in part with Federal funds from the National Institute of Allergy and Infectious Diseases, National Institutes of Health, and Department of Health and Human Services to support the Seattle Structural Genomics Center for Infectious Disease (SSGCID) under

Contract No. HHSN272201700059C (P.J.M.). Research reported in this publication was supported by the National Institute for Allergy and Infectious Disease of the National Institutes of Health under award number R01AI155536 (B.L.S.), by a fellowship from Coordination for the Improvement of Higher Education (CAPES), Brazil (STINT-PROJ-20181009558P) (D.R.H.), by the Swedish Research Council (2016-05627) and (2021-03667) (P.S. and M.G.), and by grant R21 AI 151344 from the National Institutes of Health (A.K. and E.K.K.G.).

## Author contributions

Conceptualization: B.L.S., A.K., and M.G.; Methodology: D.R.H., K.V., M.G., W.R., J.S., B.L.S., and A.K.; Investigation: D.R.H., K.V., R.Z., M.F., B.S., E.K.K.G., W.R., J.S., and B.L.S.; Visualization: D.R.H., M.F., K.V., and E.K.K.G.; Funding acquisition: P.M., B.L.S., A.K., M.G., and P.S., Project administration: B.L.S., M.G., and A.K.; Supervision: B.L.S., M.G., and A.K., Writing—original draft: D.R.H., K.V., B.L.S., M.F., M.G., and A.K.; Writing—review & editing: all authors.

## Funding

## Competing interests

The authors declare no competing interests.
