## [Peer Review File · Nature Communications]

Identification of potent and Selective N-myristoyltransferase inhibitors of Plasmodium vivax liver stage hypnozoites and schizontsREVIEWER COMMENTS

Reviewer #1 (Remarks to the Author):

With impending failure of artemisinin and only the 8-aminoquinolines being active against liver stage *P. vivax* there is an emergent need for new drugs to treat malaria that act by new mechanisms. Therefore the paper is timely.

NMT inhibitors have been developed before for malaria, but selectivity against the host has been an issue. This is the key barrier to developing an NMT inhibitor for malaria. The group effectively addresses this issue.

Combination of structure-based and pharmacophore-based approaches is state of the art and well executed.

- Initial hypotheses are well validated quickly by the approach with reasonable potency and high selectivity compounds emerging.
- Crystal structure reveals unique binding mode for the new scaffold, partially fulfilling interactions of both prior scaffolds.
- However, blood stage activities are fairly poor – 350-1250 nM EC50s vs exciting early lead being more like < 100 nM.
- There is demonstrable activity against liver stage parasites – but mostly in the micromolar EC50s. Probably real but not likely an achievable plasma concentration. All in cellular measures.
- The compounds with both reasonable potency and high selectivity are interesting (12b and 30a) and worth follow-up.

All of the conclusions drawn are well supported by the data and the compounds highlighted to represent an important step forward. However, while this is an interesting and well supported result, the work would be much more impactful with more potent compounds, ideally with stronger data, and at least some indication that an effective concentration can be reached in vivo. Ideally an animal model of activity would be included

Reviewer #2 (Remarks to the Author):

The manuscript reports the synthesis of novel Plasmodium NMT inhibitors and their activity against asexual blood stages of *P. falciparum* and hepatic stages of *P. vivax*. Plasmodium NMT has been investigated by several groups as a potential drug target. However, challenges in selectivity and the facile selection of resistant mutants have made it a less attractive target.

Rodriguez-Fernandez et al add to the body of work on Pf NMT inhibitors by reporting a new series developed using a structure-guided approach. In addition to testing compounds against *P. falciparum* asexual stages, they tested against *P. vivax* hepatic schizonts and hypnozoites.

Results are robust and the experimental methods are well described. A few missing pieces:

- 1) there is no quantification of cytotoxicity to primary human hepatocytes (this may be quite different from that of a cell line). This should be determined for the frontrunner compounds. In the absence of these data it is premature to conclude that decrease in hypnozoite/hepatic schizont numbers is entirely due to anti-parasitic effects.
- 2) On-target activity of compounds should be evaluated using myristoylation assays.
- 3) It would be helpful for the reader to see dose-response curves with the X-axis in log.
- 4) There should be an analysis of the PfNMT binding pocket. Is the binding mode identical to PvPKG?

Reviewer #3 (Remarks to the Author):

Grotli and coworkers report on the development of potent and selective N-myristoyltransferase (NMT) inhibitors of Plasmodium vivax (*P. vivax*). The motivation for the study is that the currently available NMT inhibitors have relatively poor selectivity for *P. vivax* over human NMT. Interestingly, the authors employ a structure-guided hybridization approach to create a new inhibitor that incorporates moieties from different inhibitors. Using a combination of extensive medicinal chemistry, X-ray crystallography and *P. vivax* inhibition studies, they develop compounds with IC50 values in the mid-nanomolar range and selectivities of over 100 (~270 in the best case). The authors also show that these compounds show minimal toxicity in human hepatoma cells and are active against blood stage parasites (liver stage schizonts and hypnozoites). Such inhibitors could represent new lead antimalarials.

Overall, the study seems to be rigorous and compelling, and the data is clearly described and illustrated. The X-ray crystal structure of PvNMT bound to compound 12b is of high quality and the very high-resolution structure that is afforded appears to allow for the accurate placement of key interacting

residues, particularly Try211 that allows the authors to make a reasonable argument for the molecular basis for selectivity of 12b and related compounds for PvNMT over hNMT.

My only significant criticism of the study is that it is largely a drug development study so might be more appropriate for a more specialized journal.

Some minor criticisms are listed below:

1. Line 66, deficiency⁶ needs to superscript the 6.
2. Line 81, Schizonts need to be defined.
3. Figure 1, Tyr211 is not labeled in 1b and the two side chain conformations that are described are not shown.
4. Line 136, Table 1 needs to be referenced when referencing the 'Y' and 'X' positions.
5. The authors should include a supplementary figure showing simulated annealing omit density of the inhibitor and interacting residues to help validate the modeling of the inhibitor in the active site.

Reviewer #1 (Remarks to the Author):

With impending failure of artemisinin and only the 8-aminoquinolines being active against liver stage *P. vivax* there is an emergent need for new drugs to treat malaria that act by new mechanisms. Therefore the paper is timely.

NMT inhibitors have been developed before for malaria, but selectivity against the host has been an issue. This is the key barrier to developing an NMT inhibitor for malaria. The group effectively addresses this issue.

Combination of structure-based and pharmacophore-based approaches is state of the art and well executed.

- Initial hypotheses are well validated quickly by the approach with reasonable potency and high selectivity compounds emerging.
- Crystal structure reveals unique binding mode for the new scaffold, partially fulfilling interactions of both prior scaffolds.
- However, blood stage activities are fairly poor – 350-1250 nM EC50s vs exciting early lead being more like < 100 nM.
- There is demonstrable activity against liver stage parasites – but mostly in the micromolar EC50s. Probably real but not likely an achievable plasma concentration. All in cellular measures.
- The compounds with both reasonable potency and high selectivity are interesting (12b and 30a) and worth follow-up.

All of the conclusions drawn are well supported by the data and the compounds highlighted to represent an important step forward. However, while this is an interesting and well supported result, the work would be much more impactful with more potent compounds, ideally with stronger data, and at least some indication that an effective concentration can be reached in vivo. Ideally an animal model of activity would be included.

Response: We thank this reviewer for their positive review of our manuscript; as mentioned by the editor, the animal work is beyond the scope of the work presented.

Reviewer #2 (Remarks to the Author):

The manuscript reports the synthesis of novel Plasmodium NMT inhibitors and their activity against asexual blood stages of *P. falciparum* and hepatic stages of *P. vivax*. Plasmodium NMT has been investigated by several groups as a potential drug target. However, challenges in selectivity and the facile selection of resistant mutants have made it a less attractive target. Rodriguez-Fernandez et al add to the body of work on Pf NMT inhibitors by reporting a new series developed using a structure-guided approach. In addition to testing compounds against *P. falciparum* asexual stages, they tested against *P. vivax* hepatic schizonts and hypnozoites.

Results are robust and the experimental methods are well described. A few missing pieces: 1) there is no quantification of cytotoxicity to primary human hepatocytes (this may be quite different from that of a cell line). This should be determined for the frontrunner compounds. In the absence of these data it is premature to conclude that decrease in hypnozoite/hepatic schizont numbers is entirely due to anti-parasitic effects.

Response: We thank this reviewer for their comment, and agree it is a good idea to evaluate toxicity in primary hepatocytes as well as the cell line HepG2. We have quantified the number of intact primary hepatocytes after 8 days of drug treatment in *P. vivax*-infected cultures (added as Supplementary Fig. 81). We observed minimal toxicity in primary hepatocytes, even at their highest concentration of 20 μ M. We have included the raw data in the new Supplementary File 3.

2) On-target activity of compounds should be evaluated using myristoylation assays.

Response: We agree with the reviewer that this is a critical assay – and, indeed, myristylation assays were our primary selection criteria (Tables 1 and 2, Line 422. NMT Activity Assay). To further clarify this for readers of the manuscript, we have included an additional supplemental cartoon that describes the assay we used for down-selection in Supplementary Fig. 1.

3) It would be helpful for the reader to see dose-response curves with the X-axis in log.

Response: In the revised manuscript, we have included a Supplemental File 3 with all raw data, so readers can graph the data in the format they feel is most useful for visualization.

4) There should be an analysis of the PfNMT binding pocket. Is the binding mode identical to PvPKG?

Response: We assume this reviewer is referring to PvNMT, not PvPKG. There are not currently any solved structures of PfNMT, so any analysis of this point is primarily speculation. In the revised manuscript, we have included a discussion of this point on line 368:

There are currently no structures of *P. falciparum* NMT available in the PDB, but a comparison of its amino acid sequence with that of PvNMT shows a tyrosine-to-phenylalanine substitution at the PvTyr334 site. Because **12b** is not expected to introduce a steric clash in PfNMT, this and related compounds would be valuable to consider for selective inhibition. A key question is whether PfNMT will display similar differences in conformational plasticity with HsNMTs. However, the chemical difference (change from Tyr to Phe) in the binding site of PfNMT *vis à vis* PvNMT and HsNMTs opens up the possibility for much higher selectivities beyond what is achievable through conformational selectivity alone.

Reviewer #3 (Remarks to the Author):

Grotli and coworkers report on the development of potent and selective N-myristoyltransferase (NMT) inhibitors of *Plasmodium vivax* (*P. vivax*). The motivation for the study is that the currently available NMT inhibitors have relatively poor selectivity for *P. vivax* over human NMT. Interestingly, the authors employ a structure-guided hybridization approach to create a new inhibitor that incorporates moieties from different inhibitors. Using a combination of extensive medicinal chemistry, X-ray crystallography and *P. vivax* inhibition studies, they develop compounds with IC50 values in the mid-nanomolar range and selectivities of over 100 (~270 in the best case). The authors also show that these compounds show minimal toxicity in human hepatoma cells and are active against blood stage parasites (liver stage schizonts and hypnozoites). Such inhibitors could represent new lead antimalarials.

Overall, the study seems to be rigorous and compelling, and the data is clearly described and illustrated. The X-ray crystal structure of PvNMT bound to compound **12b** is of high quality and the very high-resolution structure that is afforded appears to allow for the accurate placement of key interacting residues, particularly Try211 that allows the authors to make a reasonable argument for the molecular basis for selectivity of **12b** and related compounds for PvNMT over hNMT.

My only significant criticism of the study is that it is largely a drug development study so might be more appropriate for a more specialized journal.

Some minor criticisms are listed below:

1. Line 66, deficiency⁶ needs to superscript the 6.

Response: We have made this modification to the manuscript.

2. Line 81, Schizonts need to be defined.

Response: We have defined schizonts within the manuscript. This is now on line 61.

3. Figure 1, Tyr211 is not labeled in 1b and the two side chain conformations that are described are not shown.

Response: In the revised manuscript, we have now labeled Tyr211 in Figure 1.

4. Line 136, Table 1 needs to be referenced when referencing the 'Y' and 'X' positions.

Response: In the revised manuscript, we have added references to Table 1. This is now on line 137 and 139.

5. The authors should include a supplementary figure showing simulated annealing omit density of the inhibitor and interacting residues to help validate the modeling of the inhibitor in the active site.

Response: We thank the reviewer for this suggestion. We have added a figure to the Supporting information that addresses this reviewer's concerns. The caption reads as follows:

Supplementary Fig. 3. Inhibitor structure validation. A $2F_o-F_c$ simulated annealing composite omit map was computed using PHENIX using torsional angle dynamics^{7,8} and was displayed around **12b** using Chimera⁶ at a contour level of approximately 1.5 times the root mean square value of the map.

REVIEWERS' COMMENTS

Reviewer #2 (Remarks to the Author):

The revision addresses most of my concerns.

An outstanding issue that needs further clarification is the analysis of the PfNMT binding pocket. The authors response in lines 368-375 is difficult to understand. Homology modeling should enable generation of a high-quality model for PfNMT even in the absence of X-ray structures.

"...a comparison of its amino acid sequence with that of PvNMT shows a tyrosine-to-phenylalanine substitution at the PvTyr334 site."

> Is the Tyr334 to Phe the only change in PfNMT's binding pocket vis-à-vis PvNMT? If so then this should be clearly stated. If there are additional changes in PFNMT's binding pocket, these should be discussed

"Because 12b is not expected to introduce a steric clash in PfNMT, this and related compounds would be valuable to consider for selective inhibition."

>This sentence is a bit cryptic. Authors should elaborate on how the Tyr334Phe is relevant to steric clash by 12b and to selective inhibition of PFNMT vis-a-vis humanNMT.

Reviewer #1: No remarks

Reviewer #2 (Remarks to the Author):

The revision addresses most of my concerns.

An outstanding issue that needs further clarification is the analysis of the PfNMT binding pocket. The authors response in lines 368-375 is difficult to understand. Homology modeling should enable generation of a high-quality model for PfNMT even in the absence of X-ray structures.

"...a comparison of its amino acid sequence with that of PvNMT shows a tyrosine-to-phenylalanine substitution at the PvTyr334 site."

> Is the Tyr334 to Phe the only change in PfNMT's binding pocket vis-à-vis PvNMT? If so then this should be clearly stated. If there are additional changes in PFNMT's binding pocket, these should be discussed.

Response: Yes, UniProt accession codes Q8ILW6 and A5K1A2 (see text below).

"Because 12b is not expected to introduce a steric clash in PfNMT, this and related compounds would be valuable to consider for selective inhibition."

>This sentence is a bit cryptic. Authors should elaborate on how the Tyr334Phe is relevant to steric clash by 12b and to selective inhibition of PFNMT vis-a-vis humanNMT.

Response: We agree and do not discuss steric interactions.

The altered paragraph is incorporated into the previous paragraph, which reads as follows:

The high-resolution crystal structure of PvNMT bound to 12b demonstrates that a hybrid compound can bind within the active site. The high selectivity is likely attributed to the distinct way the core and head groups interface with the adjacent waters and polar sites near the carboxy terminus, which accompanies conformational changes that propagate throughout the peptide and myrCoA binding sites. The selectivity of compound 30a, the most selective inhibitor of the presently developed series, is also expected to leverage the adjacent water channel via insertion of its naphthalene group. These two compounds, and close derivatives, should also be considered for selective inhibition of P. falciparum NMT. There are presently no crystal structures of PfNMT available in the Protein Data Bank, but a comparison of amino acid sequences (UniProt accession codes Q8ILW6 and A5K1A2) shows only one amino acid replacement in the 12b binding site - a phenylalanine substitution at the PvNMT Tyr334 site. A key question is whether the same mechanisms leading to the high selectivities of 12b and 30a will act in the peptide binding cleft of PfNMT, and if the phenylalanine substitution can have an additive effect.

Reviewer #3 No remarks